# Cytohistological Correlation in Pleural Effusions Based on the International System for Reporting Serous Fluid Cytopathology

**DOI:** 10.3390/diagnostics11061126

**Published:** 2021-06-20

**Authors:** Daniel Pinto, Eduardo Cruz, Diamantina Branco, Cláudia Linares, Conceição Carvalho, Amélia Silva, Martinha Chorão, Fernando Schmitt

**Affiliations:** 1Serviço de Anatomia Patológica, Centro Hospitalar de Lisboa Ocidental, EPE, 1349-019 Lisboa, Portugal; danielgomespinto@gmail.com (D.P.); ecruz@chlo.min-saude.pt (E.C.); dmvasconcelos@chlo.min-saude.pt (D.B.); clinares@chlo.min-saude.pt (C.L.); mcbcarvalho@chlo.min-saude.pt (C.C.); mabsilva@chlo.min-saude.pt (A.S.); mchorao@chlo.min-saude.pt (M.C.); 2NOVA Medical School, 1169-056 Lisboa, Portugal; 3IPATIMUP-Instituto de Patologia e Imunologia Molecular, Universidade do Porto, 4200-135 Porto, Portugal; 4Departamento de Patologia, Faculdade de Medicina da Universidade do Porto, 4200-135 Porto, Portugal; 5RISE@CINTESIS, 4200-450 Porto, Portugal

**Keywords:** serous, effusion, international system, molecular pathology, cytology, standardization

## Abstract

The International System for Reporting Serous Fluid Cytology (TIS) was recently developed. Given its novelty, most studies looking into the risk of malignancy (ROM) of serous effusion diagnostic categories were published before the development of TIS. We searched the database of our department for pleural effusions diagnosed in the last five years, excluding those without a corresponding pleural biopsy. Cases were reviewed and reclassified according to the TIS. A cytohistological correlation was performed. In total, 350 pleural effusion specimens with one or more corresponding pleural biopsies were included. After reclassification, 5 (1.43%) were nondiagnostic (ND), 253 (72.29%) were negative for malignancy (NFM), 7 (2.00%) had atypia of unknown significance (AUS), 14 (4.00%) were suspicious for malignancy (SFM), and 71 (20.57%) were malignant (MAL). Calculated ROM was 40% for ND, 20.16% for NFM, 42.86% for AUS, 78.57% for SFM, and 100% for MAL. Effusion cytology sensitivity and specificity were 60.29% and 98.56%, respectively. This is the first publication looking into the cytohistological correlation of a retrospective cohort of pleural effusions based on the TIS. We add to the body of data regarding the ROM for TIS categories, highlighting areas of potential future research.

## 1. Introduction

Serous effusions may develop in the setting of both neoplastic and non-neoplastic conditions. Effusion specimens are easily accessible with minimal complications, and examination of these specimens may yield important diagnostic information, as well as provide adequate material for ancillary diagnostic testing and the increasingly important molecular theranostic tests [1,2].

Pleural effusions, in particular, may be secondary to both primary neoplasms, such as malignant mesothelioma (MM) or primary effusion lymphomas, and secondary metastatic malignancies, most often from the breast and lung [3].

Patients presenting with neoplastic pleural effusions are often severely ill and in need of urgent care. These harbor a dismal prognosis, and as such a prompt therapeutic intervention is paramount. Effusion specimens, when correctly handled and examined, may enable a fast and reliable diagnosis, with a profound impact on clinical management [4,5,6].

Unfortunately, until very recently, there was no standardized system or terminology to guide serous effusion cytology diagnoses. Therefore, different laboratories have adopted different diagnostic terms and have relied on different criteria and sample management. This has led to widely varying results between institutions with effusion specimens, lowering trust from clinicians and hampering clinical research [7,8].

Following in the footsteps of other nomenclature systems for reporting cytology, the International Academy of Cytology (IAC) and American Society of Cytopathology (ASC) sponsored a workforce of experts to create a standardized system for serous effusion cytology reporting named the International System for Reporting Serous Fluid Cytology (TIS). Based on the most up-to-date literature and expert consensus, TIS aims to serve as a framework through which reporting variability may be reduced [9,10,11,12,13,14,15].

Furthermore, TIS defines clear diagnostic criteria for each category. This is expected to improve the interobserver agreement and provide well-defined risks of malignancy (ROM) for each category and improve patient management [16]. 

The five diagnostic categories defined by TIS are nondiagnostic (ND), negative for malignancy (NFM), atypia of undetermined significance (AUS), suspicious for malignancy (SFM), and malignant (MAL) [16]. 

Several recent publications have looked at the risk of malignancy (ROM) for each of these categories. They have shown that values vary between affected cavities [5,6,17,18,19,20,21,22]. Most studies, however, were published before the development and implementation of TIS, with few exceptions [5,6,17,18,19,20,21,22].

In this paper, we aim to add to the body of data regarding the ROM in pleural effusions by retrospectively reclassifying a case series according to TIS and performing cyto-histological correlation.

## 2. Materials and Methods

### 2.1. Case Series and Data Gathering

The database of our department was searched for all pleural effusions diagnosed between January 2016 and March 2021. From these, we excluded all cases without a concomitant or follow-up pleural biopsy. Clinical files of cases with follow-up biopsies were reviewed, when appropriate, to exclude cases with etiologies unrelated to biopsy findings.

Cytology reports were then reviewed for each case. The following data were gathered from each report:Submitted sample volume;Microscopic descriptions;The diagnosis that was performed and based only on the morphology of the cytology slides;Accompanying cellblock (if any) and cellblock diagnosis;Ancillary testing, which was performed directly on effusion material cytology slides.

According to the gathered information, all cases were then reclassified according to the criteria defined in the TIS for each of the five aforementioned categories [16]. 

Summarily, these include the following:ND: Specimens providing no diagnostic information;NFM: Specimens comprised of mostly mesothelial cells without cytological atypia;AUS: Specimens with atypical cells but which lack either the criteria or cellularity to be included in the SFM or MAL categories. A diagnosis of AUS favors benignity;SFM: Specimens with atypical cells that strongly raise the suspicion of malignancy, but whose observation is limited by low cellularity of artifacts. A diagnosis of SFM favors malignancy;MAL: Specimens with overt features of malignancy, either primary or secondary.

When the information contained in the report was considered insufficient, the original slides were reviewed by a single cytopathologist (DP) and classified in the best fitting TIS category. Furthermore, since in our practice the AUS category has historically not been used, all negative cases with cellblocks were reviewed, regardless of the information contained in the reports.

In our laboratory, all effusion cytology specimens during this period were processed and prepared as sediments, either using the cytospin technique or using ThinPrep liquid-based cytology. Effusion samples are usually submitted to the laboratory fresh, although from March 2020 to March 2021, they were mostly submitted fixated in alcohol at or above a concentration of 70% due to coronavirus disease 2019 (COVID-19) concerns [23]. 

Additionally, mostly for logistical reasons, we do not send material from effusion samples for flow cytometry analysis, even if submitted fresh—this procedure is conducted directly by the clinician gathering the sample. 

The reports of concomitant or follow-up pleural effusion biopsies were also gathered, and the final diagnoses made on these specimens were recorded. When the biopsy was insufficient for diagnosis, when there were discordant findings with cytology samples, or when otherwise considered appropriate, clinical records were reviewed, including imaging findings, to determine the final patient diagnosis.

### 2.2. Data Entry and Statistical Analysis

All data were inserted in a Office Excel (Microsoft, Redmond, WA, USA) spreadsheet, each line representing a pleural effusion cytology sample. Diagnoses performed based on cytological morphology alone were considered provisional if they either had an accompanying cellblock or immunohistochemistry (IHC) performed on cytology slides. In these specimens, diagnoses including information cellblocks and IHC were considered the definitive cytological diagnosis. For each cytology specimen, a corresponding concomitant or follow-up pleural biopsy was identified, and its diagnosis registered in the same Excel spreadsheet line. In some cases, different effusion samples from a single patient were attributed to the same corresponding biopsy. If more than one biopsy specimen was available for a given patient, during the predefined time period, nondiagnostic or negative biopsies were discarded in favor of a positive result. Biopsy diagnoses were considered final if prior cytology samples were negative, or if there was an agreement between the diagnoses performed in different sample types. In discordant or dubious cases, clinical and imaging findings were reviewed to arrive at a final diagnosis. This final diagnosis was considered the gold standard for the calculation of ROM and performance analysis. These values were calculated for the definitive cytology diagnosis and for pleural biopsies. Furthermore, and following the methodology proposed by Lobo et al., performance analysis was carried out considering MAL as positive, MAL and SFM as positive, and MAL, SFM, and AUS as positive. Continuous variables were represented as ranges, medians, and means. Distribution of continuous variable by year or TIS diagnostic category was analyzed by one-way analysis of variance (ANOVA) tests, performed using GraphPad Prism 6 (GraphPad Software Inc., San Diego, CA, USA). *p*-values of <0.05 were considered statistically significant.

## 3. Results

### 3.1. Series Description

Between January 2016 and March 2021, a total of 1228 pleural effusion specimens were diagnosed in our department. From these, a total of 350 effusion specimens had one or more concomitant or follow-up pleural biopsies. In all, our series included 169 pleural biopsies from 135 different patients. 

Of those included, 72 (53.33%) patients were male and 63 (46.67%) were female (male to female ratio of 1.14:1). The median age was 76 years (mean 71.63 years; range 20–94 years). The median number of pleural effusion samples per patient was two (mean 2.59; range 1–7). Thirty patients had two pleural biopsies performed at different times and two patients had three. Most biopsies were performed either at the same time as the collection of the effusion specimen or within one month. Of those that were performed after this time interval, after review of the clinical findings, none showed alternative etiologies differing from the biopsy specimen findings, and thus, no further cases were excluded. 

A summary of the case series and specimen characteristics can be found in Table 1.

### 3.2. Cytological Diagnosis

After review according to the TIS, from the 350 included pleural effusion specimens and based on the morphological findings of the cytology slides alone, 7 (2.00%) were classified as ND, 223 (63.71%) were classified as NFM, 35 (10.00%) were classified as AUS, 30 (8.57%) were classified as SFM, and 55 (15.71%) were classified as MAL. A total of 78 specimens had ancillary immunohistochemistry (IHC) testing performed, 20 directly on cytology slides only, 45 on cellblocks alone, and 13 specimens on both cytology slides and cellblocks. IHC provided adequate information to change the diagnosis in 47 cases. Thus, including this information, 2 cases initially classified as ND were reclassified as NFM; 28 cases initially classified as AUS were reclassified, 27 as NFM and 1 as MAL; 16 cases initially classified as SFM were reclassified, 1 as NFM and 15 as MAL. An example of one of these cases can be seen in Figure 1. No NFM or MAL cases were reclassified when IHC was considered.

This led to a definite cytological classification of 5 (1.43%) cases as ND, 253 (72.29%) cases as NFM, 7 (2.00%) cases as AUS, 14 (4.00%) cases as SFM, and 71 (20.29%) cases as MAL.

Comparative distribution of cases at each classification round can be found in Table 2.

From the 350 pleural effusion specimens analyzed, 344 had information regarding the total specimen volume. In this series, a median of 25 mL of fluid was submitted to the pathology laboratory (mean 31.60 mL; range 2–300 mL). There were no statistically significant differences found between the TIS classification or year of sample collection in terms of specimen volume (*p* < 0.005). 

### 3.3. Histologic Diagnosis, Clinical Data, and Other Ancillary Methods

From the 169 pleural biopsies, 5 (2.96%) provided insufficient material for diagnosis, 114 (67.46%) were negative for malignancy, and 50 (29.59%) were positive for malignancy. One of the patients with insufficient material for diagnosis on biopsy material was diagnosed with a diffuse large B-cell lymphoma on a cellblock from a concomitant effusion specimen. Two others, also with insufficient tissue for diagnosis, were diagnosed in concomitant effusion specimens using flow cytometry, one with follicular lymphoma and the other with a small lymphocytic lymphoma. In addition, 12 other patients, with a biopsy without evidence of malignancy, ended up being diagnosed with a malignant neoplasm through other methods. Six of them had positive concomitant pleural effusions showing metastatic carcinomas from the breast and lung, characterized in cellblocks. The remaining six were diagnosed with a malignant effusion through clinical and imagiological correlation. Four had metastatic carcinomas from the breast and lung and two had malignant mesothelioma. In these cases, histological confirmation of the diagnosis was obtained through biopsies from other topographies, including the bone and mediastinal lymph nodes.

### 3.4. Risk of Malignancy and Performance Analysis

Integrating the histological and cytological diagnoses with clinical data and other ancillary diagnostic tests performed, from the 135 patients, 58 (42.96%) had a final diagnosis of malignancy and 77 (57.04%) were found to have effusions from other, non-neoplastic causes. This was considered the gold standard for performance analysis. In our series, the most frequent malignancies diagnosed were, in decreasing order of frequency, Adenocarcinoma from the lung, Carcinoma of nonspecial type from the breast, and malignant mesothelioma. An illustrative case of malignant mesothelioma can be seen in Figure 2. A complete list of final diagnoses can be found in Table 3.

In cytology specimens, the ROM for ND was 40%, 20.16% for NFM, 42.86% for AUS, 78.57% for SFM, and 100% for MAL. Considering MAL and SFM diagnoses as positives, sensitivity was 60.29%, specificity was 98.56%, positive predictive value (PPV) was 96.47% and negative predictive value (NPV) was 79.23%.

In the case of 17 patients, the final diagnosis was different from the one obtained through pleural biopsy alone. Thus, in our case series, pleural biopsies by themselves showed a sensitivity of 78.13%, a specificity of 100%, a PPV of 100%, and an NPV of 87.72%.

The summary of performance analysis can be found in Table 4. 

## 4. Discussion

Serous effusions develop in the setting of both neoplastic and non-neoplastic conditions. When a malignancy is suspected, effusion draining is often the first diagnostic step, given that they are easily accessible and obtained with minimal complications.

As we have previously discussed, before the publication of TIS, each laboratory and practice had no choice but to establish their own diagnostic categories and criteria. This has led to heterogeneity in the diagnostic yield obtained from effusion specimens. Given that TIS is a recent publication, there is still a dearth of data regarding its implementation.

However, several retrospective case series and reviews of note have been published on the subject. 

In a seminal 2019 publication, Farahani et al. performed a meta-analysis of 80 studies, including 39941 effusion samples. Despite being published before the TIS, the terminology used closely approximates that of the international system TIS. Authors could not account for the varying criteria between publications, however. Polling all samples together, from different topographies, including the pleura, the authors arrived at a combined ROM for ND of 17.4% ± 8.9% (range 0–100%); 20.7% ± 0.3% (range 0–81%) for NFM; 65.9% ± 10.6% (range 13.3–100%) for AUS; 81.8% ± 4.8% (range 33.3–100%) for SFM, and 98.9% ± 0.1% (range 87–100%) for MAL. Additionally, they reported a sensitivity and specificity in pleural effusions of 73.2% and 99.9%, respectively [6]. 

In this same year, Valerio et al. reported on a series of 65 pleural effusions with matching biopsies and found a ROM of 50% for ND, 44% for NFM, 50% for AUS, 83.3% for SFM, and 96.2% for MAL. They reported a sensitivity of 69.4%, a specificity of 93.3%, a PPV of 96.2%, and an NPV of 56% [24]. 

In 2020, Lobo et al. published a series of cytohistological correlations in effusion fluids from various topographies. The authors found a ROM of 57.1% for ND, 23.9% for NFM, 50% for AUS, 76.2 for SFM, and 100% for MAL in a series of 628 pleural effusions with histological correlation. Considering MAL and SFM diagnoses as positive, they reported a sensitivity of 66.9%, a specificity of 98.4%, a PPV of 97.6%, and an NPV of 75.8% in pleural effusions [17]. 

In one of the first publications on the topic since the publication of TIS, Hou et al. found a ROM of 39% for the AUS category and 64% for the SFM category in a series of 145 serous effusions, including 91 pleural effusions. No other diagnostic categories were evaluated in this publication. Sensitivity and specificity values were not calculated [22].

Our results are in line with this bibliography and show the diagnostic value that can be obtained through the examination of these specimens.

In our series, the calculated ROM for ND was 40%, 20.16% for NFM, 42.86% for AUS, 76.92% for SFM, and 100% for MAL. Considering MAL and SFM as positives, sensitivity was 59.56%, specificity was 98.56%, PPV was 96.43%, and NPV was 78.93%.

It is apparent that the ROM associated with the ND category varies widely between publications. In our series, we only had a residual number of cases with an ND diagnosis (N = 5; 1.43%) and thus consider this result unreliable. This is in line with previously published literature and highlights the fact that most effusion samples harbor enough cellularity for morphological analysis, which coincidentally leads to methodological difficulties in determining the true ROM value for this category. Interestingly, in our series, all five ND cases had subsequent effusions with adequate cellularity for diagnosis. Therefore, if clinically appropriate, we believe pleural effusion samples should be repeated until a diagnostic one is obtained.

The values we report for NFM, AUS, and SFM overlap with the literature reviewed above. It is noteworthy that the values reported by us, Lobo et al., and Hao et al. for the AUS category were lower than what had been previously reported by Farahani et al. and Valerio et al. This might hint at an effect of the better-defined TIS criteria for this indeterminate category, placing it closer to NFM.

Our performance analysis is also in agreement with previous publications. However, the sensitivity value we found is slightly lower than what had previously been reported. Taking a closer look at the data, we can infer that our series has a relatively large number of malignant mesotheliomas (*n* = 24; 17.27% of all malignant cases), 13 of which were classified as NFM and 1 as AUS. These cases were signed out as reactive mesothelial hyperplasia and atypical mesothelial hyperplasia, respectively. Given that this is a retrospective study, the results are easy to understand: historically, the diagnosis of malignant mesothelioma on effusion specimens has been a contentious issue. Furthermore, it is only recently that immunohistochemistry that enables a reliable differential diagnosis between this malignancy and reactive mesothelial hyperplasia has become available [25]. Thus, most of these cases were not submitted for cellblock preparation. Therefore, in the context of malignant mesothelioma, a diagnosis of SFM or MAL was made only in the right clinical context and when morphological features were overtly exuberant, leading to the observed high rate of false negatives. See Figure 2. 

Furthermore, in our series, we had four cases of small cell carcinomas of the lung and three of poorly cohesive carcinomas of the stomach, which were classified as NFM. These diagnoses are easily missed when cellularity is low, and on review of the glass slides, no criteria were found that would allow for recategorization into AUS, SFM, or MAL.

Thus, and given that specificity, PPV and NPV are close to what has previously been reported; this slight deviation can be construed as deriving from the specific characteristics of our series and inherent to its retrospective nature.

Still, on the topic of performance analysis, we would like to highlight that our results hint that, for this purpose and perhaps for clinical management, it is best to consider the categories of MAL and SFM as positives, while excluding AUS. This makes sense from the perspective of TIS and is supported in our series by better overall sensitivity and NPV with a reduced cost to specificity and PPV.

Another interesting finding of our case series relates to volume. The median effusion volume submitted for analysis was low, ranging between 20 and 60 mL between TIS categories. All TIS categories showed an overlap in submitted volume, including the ND category, and there was no statistically significant difference in fluid distribution between diagnostic categories. Despite the low volume of submitted samples, we had a low number of ND cases, in line with what has previously been reported in the literature. This is interesting, particularly in light of other studies that have suggested that a minimum of 50–75 mL of fluid should be submitted to ensure enough material for diagnosis [26,27,28,29,30,31]. We failed to replicate this finding. In fact, many of our diagnostic cases had a very low volume submitted to the laboratory, apparently without a detrimental effect on diagnostic yield. 

Our study has a few limitations, which should be addressed. Firstly, despite the size of the series, a significant number of cases come from the same patients. This leads to certain peculiarities, such as the high prevalence of malignant mesothelioma cases discussed above. However, it is an unavoidable part of our methodology, as it derives from the fact that only cases containing an accompanying pleural biopsy were included. Additionally, the fact that only cases including a pleural biopsy were included is a bias in itself: pleural biopsies are not conducted routinely but are instead performed in more clinically alarming cases. This may lead to overestimated values for ROM, particularly in the NFM and AUS categories. However, this is a limitation of retrospective cytohistological correlation studies that can only be overcome in prospective, clinical trials and affects many of the previous publications on this subject. Furthermore, our institution historically did not use the AUS category. We tried to overcome this limitation by reviewing all cases with cellblocks as well as those in which the information in the reports was not clear, or if any doubts remained, for whatever reason. We also do not send fluid directly for flow cytometry analysis. In this series, this turned out not to be a problem, due to close articulation with the clinical departments and ease of obtaining a repeat specimen for this purpose when a lymphoid malignancy is suspected. Three of the five cases harboring lymphoid malignancies were submitted to flow cytometry, with the remaining two being characterized in cellblocks. 

Our study also has its strengths. The series size is relatively large for a cytohistological correlation study (*n* = 350). Our institution is a district hospital, treating a wide variety of pathologies, both malignant and benign. This has led to a wide variety of malignancies diagnosed in pleural effusion specimens. Submitted effusion volume was available from most samples. There were no false positives in our series. Furthermore, we have extensive experience in both cellblock preparation and in performing immunohistochemistry directly in cytology slides. Overall, 78 specimens had immunohistochemistry testing performed in our series, 20 directly only on cytology slides, 45 on cellblocks alone, and 13 specimens on both cytology slides and cellblocks.

It is noteworthy that cellblocks enabled two specimens considered ND in the cytology slides to be reclassified as NFM, although one of them later proved to be positive for malignancy in the corresponding biopsy specimen. In addition, 28 cases of AUS were reclassified because of IHC, 27 as NFM and 1 as MAL, and 20 of them in agreement with the biopsy specimen. An additional 16 SFM cases were reclassified, 1 into NFM and 15 into MAL, all correctly. Our series highlights both the usefulness and limitations of immunohistochemistry performed directly on cytology specimens. On the one hand, it allows for a quick confirmation or exclusion of a suspected morphological diagnosis even if there is no material available for cellblock preparation. On the other hand, further characterization is often needed, and this led to a subsequent cellblock request in 13 out of 20 cases with IHC performed of cytology slides. Both specimen types may be useful for the purpose of theranostic testing [16]. We would like to highlight two cases from our series of Adenocarcinoma of the lung in which cytology slides and cellblocks were the only available samples for theranostic testing, which enabled the evaluation of PD-L1 expression and molecular analysis by next-generation sequencing (NGS).

Liquid-based cytology enables a ready preparation of cellblocks for ancillary techniques and is thought to provide better diagnostic yield than conventional smears [16,32,33,34,35]. Given this evidence, we migrated to this technique in early 2020. However, in this series, we did not observe a significant difference between sample types, in terms of ROM, test performance, or sample availability for cellblock (data not shown); this can probably be explained due to an extensive experience in our institution with cytospin, accompanied by careful sample management. However, we believe our numbers are too small to draw any significant conclusions in this regard.

## 5. Conclusions

This is the first publication looking into the cytohistological correlation of a retrospective cohort of pleural effusions based on the recently published TIS. Our data show that effusion cytology is a sensitive and specific method for the diagnosis of malignancies of several types, in line with previous publications on the subject. We add to the body of published literature regarding the ROM of the different categories of the TIS. In this regard, we found some disagreement with previous publications in the intermediate categories, showing that ROM values for these still need to be better defined. We expect this to be an active area of research in the future, leading to better-defined ROMs and improved patient management using the TIS, as further studies are needed to validate our findings, namely, prospective clinical trials focused on cytohistological correlations. 

## Figures and Tables

**Figure 1 diagnostics-11-01126-f001:**
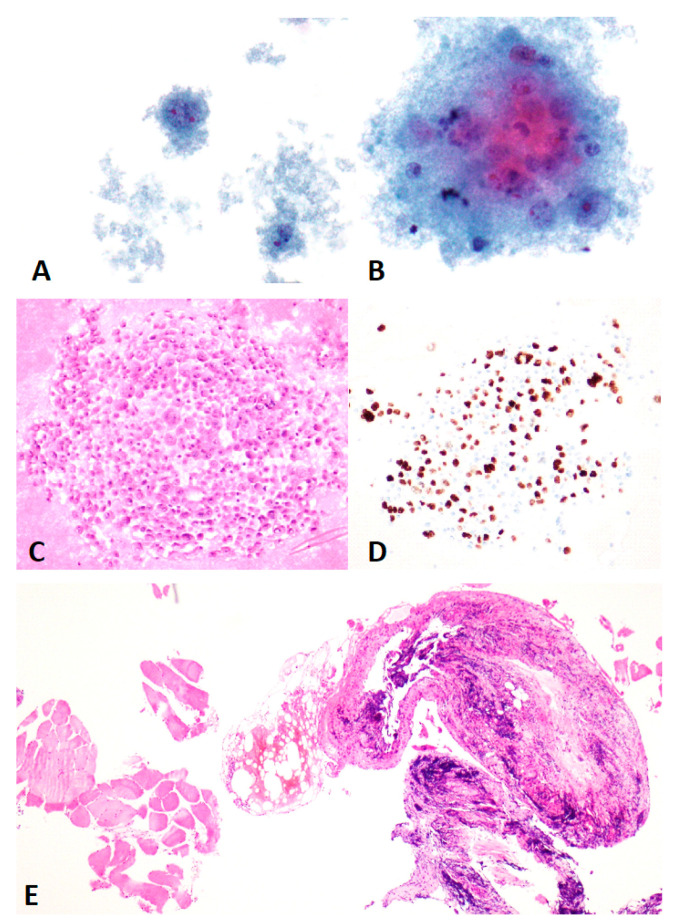
ThinPrep ((**A**) (400×) and (**B**) (600×)–Papanicolaou) with very low cellularity showing pleomorphic cells with prominent nuclei, obscured by a heavily proteinaceous background (alcohol fixation artifact), provisionally classified as suspicious for malignancy (SFM). Cellblock ((**C**) (200×)–Hematoxylin and Eosin) showed much higher cellularity and enabled a definite diagnosis of malignancy (MAL) and specifically an etiological diagnosis of adenocarcinoma from the lung, using immunohistochemistry (pictured TTF1–(**D**) (200×)). Pleural biopsy ((**E**) (40×)–Hematoxylin and Eosin) did not provide adequate material for diagnosis.

**Figure 2 diagnostics-11-01126-f002:**
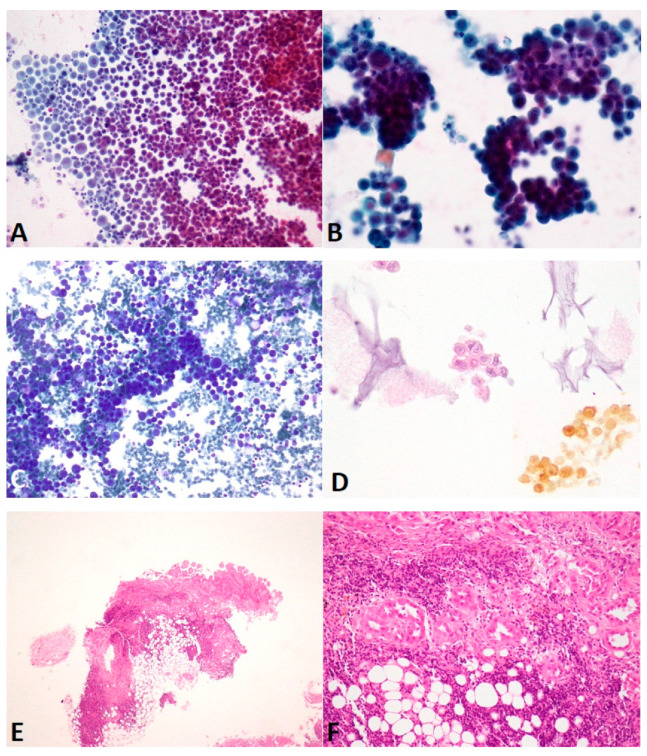
Cytospin ((**A**) (100×), (**B**) (600×), and (**C**) (100×)–Papanicolaou and May–Grunwald–Giemsa) of a case of malignant mesothelioma diagnosed as suspicious for malignancy (SFM). Notice the high number of enlarged mesothelial cells with a metachromatic cytoplasm, prominent nuclei, subtle atypia, and some binucleated forms. Morular and papillary aggregates can focally be observed ((**B**) (600×) and (**C**) (100×)). Cellblock) (**D**) (400×)–Hematoxylin and Eosin) had low cellularity but enabled confirmation of the mesothelial nature of the cells (Calretinin–inset). The diagnosis of SFM was maintained. Pleural biopsy ((**E**) (20×) and (**F**) (200×)–Hematoxylin and Eosin) confirmed the presence of malignancy, showing epithelioid malignant mesothelioma with an invasion of pleural fat.

**Table 1 diagnostics-11-01126-t001:** Patient Demographics and Specimen characteristics by TIS category.

		ND	NFM	AUS	SFM	MAL
**Number of:**	Cases	5	253	7	14	71
	Patients	4	110	7	14	37
**Gender**	Males	2	61	6	8	19
	Females	2	49	1	6	18
**Age**	Median	87	77	80	75.5	70
	Range	65–88	20–94	51–87	53–88	46–93
**Fluid volume**	Median	60	20	40	35	30
	Range	40–90	2–300	8–60	8–80	5–90
**Cellblocks**	Total	0	27	3	4	24
**Immunohistochemsitry perfromed directly on cytology slides**	Total	0	10	3	3	17
**Flow cytometry**	Total	0	0	0	2	1

**Table 2 diagnostics-11-01126-t002:** Distribution of provisional and definitive cytological diagnoses, as well as final integrated diagnosis, for each effusion sample.

	ND		NFM		AUS		SFM		MAL	
	N	%	N	%	N	%	N	%	N	%
**Provisional cytology diagnosis Excluding ancillary testing**	7	2.00	223	63.71	35	10.00	30	8.57	55	15.71
**Final cytology diagnosis** **Including ancillary testing**	5	1.43	253	72.29	7	2.00	14	4.00	71	20.29
**Final integrated diagnosis** **Including cytology, histology, and clinical/other information**	NA		212	60.57	NA		NA		138	39.43

**Table 3 diagnostics-11-01126-t003:** Final neoplastic diagnoses per TIS diagnostic category in effusion samples.

			ND	NFM	AUS	SFM	MAL
Primary	**Pleura**	**Malignant Mesothelioma (27)**	1	15	1	3	7
Secondary	**Lung**	**Adenocarcinoma (47)**	0	9	0	2	36
		**Small cell carcinoma (4)**	0	4	0	0	0
	**Stomach**	**Adenocarcinoma, NOS (4)**	0	1	0	0	3
		**Poorly cohesive carcinoma (9)**	0	3	0	1	5
	**Pancreas**	**Ductal adenocarcinoma (8)**	0	4	0	0	4
		**MiNEN (2)**	0	1	0	0	1
	**Urinary tract**	**Renal cell carcinoma (4)**	0	3	0	1	0
		**Urothelial carcinoma (5)**	0	1	1	0	3
	**Breast**	**Carcinoma, NST (17)**	0	7	0	1	9
	**Unknown primary**	**Carcinomas (6)**	1	4	1	0	0
	**Hematolymphoid**	**Small lymphocytic lymphoma (1)**	0	0	0	1	0
		**Follicular lymphoma (2)**	0	0	0	1	1
		**Diffuse large B-cell lymphoma (1)**	0	0	0	0	1
		**Primary effusion lymphoma (1)**	0	0	0	0	1

**Table 4 diagnostics-11-01126-t004:** Performance analysis in pleural effusion cytology and biopsy specimens.

Positive:	MAL	MAL + SFM	MAL + SFM + AUS	Biopsy
**Sensitivity**	52.21%	60.29%	62.50%	78.13%
**Specificity**	100.00%	98.56%	96.65%	100.00%
**PPV**	100.00%	96.47%	92.39%	100.00%
**NPV**	76.28%	79.23%	79.84%	87.72%

## Data Availability

Data available on request due to institutional restrictions related to patient privacy.

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
