# Peer review of "Cytohistological Correlation in Pleural Effusions Based on the International System for Reporting Serous Fluid Cytopathology"

_diagnostics, 2021, doi:10.3390/diagnostics11061126_

Round 1

Reviewer 1 Report

In the manuscript, the authors present a study regarding cytohistological correlation in pleural effusions based on the international system for reporting serous fluid cytopathology. The authors introduced in the study 350 pleural effusion specimens obtained from 135 patients. The general idea of the manuscript is quite interesting, but in my opinion this manuscript can not be published in this form. My observations are :

  1. In the lines 140-141 of the manuscript, the authors stated that they studied 350 pleural effusions specimens had one or more concomitant or follow-up pleural biopsy. I think that is not correct to introduce in the study cases with concomitant and follow-up biopsies. If the follow up biopsy is performed after a long interval of time from the time when the pleural effusion specimen was obtained is possible to obtain contradictory results. (the pleural effusion specimen can be benign and the pleural biopsy can be malignant in neoplastic patients).
  2. In the statistical analysis part of the manuscript, the authors introduced some information that are not related to the statistical analysis.
  3. Is not correct to introduce figures in the discussions part of the manuscript.
  4. The conclusions part of the manuscript are too long and are different from the conclusions part of the abstract.

Author Response

In the manuscript, the authors present a study regarding cytohistological correlation in pleural effusions based on the international system for reporting serous fluid cytopathology. The authors introduced in the study 350 pleural effusion specimens obtained from 135 patients. The general idea of the manuscript is quite interesting, but in my opinion this manuscript can not be published in this form. My observations are:

  1. In the lines 140-141 of the manuscript, the authors stated that they studied 350 pleural effusions specimens had one or more concomitant or follow-up pleural biopsy. I think that is not correct to introduce in the study cases with concomitant and follow-up biopsies. If the follow up biopsy is performed after a long interval of time from the time when the pleural effusion specimen was obtained is possible to obtain contradictory results. (the pleural effusion specimen can be benign and the pleural biopsy can be malignant in neoplastic patients).

Point-to-point response:

We thank the reviewer for this observation. We agree that the inclusion of follow up biopsies performed after a long interval of time can interfere with the establishment of a precise cyto-histological correlation. However, in our study most of the biopsies were performed within one month of obtaining the serous effusion sample. When this was not the case, we performed a thorough review of the clinical files, as highlighted in the methods section, to rule out other causes of effusion. We updated our methods and results to more clearly explain this.

In the statistical analysis part of the manuscript, the authors introduced some information that are not related to the statistical analysis.

Point-to-point response:

We appreciate this relevant comment. We have changed the heading of the section to “Data entry and statistical analysis” to better reflect its content.

Is not correct to introduce figures in the discussions part of the manuscript.

Point-to-point response:

Thank you for your comment. The image was taken to illustrate what we felt was an important point in the discussion. However, we referenced it earlier in the results section and called back to it during the relevant part of the discussion.

The conclusions part of the manuscript are too long and are different from the conclusions part of the abstract.

Point-to-point response:

Thank you for your valuable input. We surmised the conclusion, tried to make it more in line with our stated aims and were able to reduce the section from a total of 227 to 136 words. We also attempted to homogenize its content with the abstract.

Reviewer 2 Report

The authors evaluate a new system for reporting serous fluid cytology of pleural effusions in correlation with histology. 

The study is well conducted and informative and adds important value to the community of cytopathologists and histopathologists. 

Author Response

We would like to thank you very much for your review and kind comment. We hope to keep contributing in the future in this area of knowledge that is very dear to us.

Round 2

Reviewer 1 Report

The manuscript has been reviewed before and the authors changed the manuscript according to the previous reviewers indications. The quality of the manuscript has been improved. That is why I think that this manuscript can be published.